# Latent class analysis of IPOs in the Nordics

**Mikael Bask** *, **Anton Läck Nätter**

Department of Economics, Uppsala University, Uppsala, Sweden

* mikael.bask@nek.uu.se

## Abstract

We examine how the offer size of initial public offerings (IPOs) and the market return on their issue date are related to the pricing of 314 IPOs issued by firms in Denmark, Finland, Norway and Sweden at the one-day, one-week and four-week horizons using latent class analysis, which is a structural equation methodology. We identify four latent classes at each time horizon, where classes (i)-(ii) include a greater number of IPOs: (i) large-sized and underpriced IPOs; (ii) small-sized and overpriced IPOs; (iii) small-sized and severely underpriced IPOs; and (iv) large-sized IPOs that are overpriced at the one-day horizon but underpriced at the four-week horizon. The market returns are normal in latent classes (i)-(iii) and weak in class (iv). Approximately half of the IPOs in the technology sector are in the latent class with small-sized and overpriced IPOs, and most of the IPOs in the class with small-sized and severely underpriced IPOs are in the healthcare sector. Finally, the underpricing of IPOs is not corrected after one or four weeks of trading. Instead, the mean return and the standard deviation of returns increase with the time horizon.

**Data Availability Statement:** All relevant data are within the manuscript and its Supporting Information files.

**Funding:** The authors received no specific funding for this work.

## Introduction

When a firm goes public, the equity sold in the initial public offering (IPO) tends to be underpriced, resulting in a large increase in the stock price on the first day of trading. The magnitude of the underpricing of IPOs varies both between and within countries and over time [1] but is on average quite large (see Jay Ritter's IPO data at https://site.warrington.ufl.edu/ritter/ipo-data/). The average first-day return in the U.S. during 1960–2020 was 17.2 percent, the average first-day return in the U.K. during 1959–2016 was 15.8 percent, and the average first-day return in Japan during 1970–2020 was 48.8 percent; in contrast, the average first-day returns in China and India were as large as 170.2 (1990–2020) and 84.0 percent (1990–2020), respectively. See Loughran et al. [2] for average initial stock returns in 54 countries and Ljungqvist's [3] review of theories explaining the underpricing of IPOs.

The Nordic IPO market is, in international comparison, a small market. To provide some numbers, the underpricing of IPOs in the Nordic countries (or the Nordics) has been lower in magnitude than in the U.S., the U.K. and Japan, with the exception of Sweden, which saw a 25.9 percent average first-day return (1980–2015). The first-day returns in Denmark, Finland and Norway were 7.4 (1984–2017), 14.2 (1971–2018) and 6.7 percent (1984–2018), respectively.

**Competing interests:** The authors have declared that no competing interests exist.

The aim of this paper is to add knowledge on the pricing of IPOs in the Nordics. We do this by first extending the time horizon from the one-day horizon to the one-week and four-week horizons. This allows us to explore whether the average underpricing, or overpricing, of IPOs is corrected after one or four weeks of trading. Second, we examine how the offer size of IPOs —that is, the number of issued shares in an IPO times the offer price for those shares—and the market return on the issue date of IPOs are related to their under- or overpricing.

The intermediate and long-run performance in the pricing of IPOs are relatively under-researched. Examples of research that study the long-run pricing performance of IPOs include the seminal paper by Schultz [4], the papers on the Nordic markets by Hahl et al. [5] and Westerholm [6], and the meta-analysis by Engelen et al. [7] using a sample of 123 empirical studies. (The latter paper contains an extensive list of references for research on the pricing of IPOs but none of those studies cover the Nordic IPO market.) The time horizon in those studies is typically measured in months or years, whereas the time horizon in the present paper is measured in weeks. Hence, the research provided in this paper adds to the literature on the intermediate-run performance in the pricing of IPOs.

If market-wide news reaches the stock market on an IPO's issue date, the news will affect not only the market return and the first-day return on the issuing firm's securities but also possibly its first-week and four-week returns. The dot-com bubble, characterized by soaring stock prices accompanied by a dramatic increase in the underpricing of IPOs, supports the hypothesis of a relationship between the underpricing of IPOs and market returns [8]. We are of two minds regarding the potential relationship between the offer size and the pricing of IPOs. On the one hand, it is easy to give examples of large-sized IPOs that have been underpriced (cf. Google's IPO); on the other hand, there are many examples of small attention-grabbing IPOs that have also been underpriced [9].

Latent class analysis (LCA) is used to examine the relationship between the offer size of an IPO, the market return on the IPO's issue date and the pricing of the IPO. LCA is useful when it is suspected that groups of IPOs exist in the sample with different properties but it is not easy to identify those groups [10]. LCA identifies the groups—or latent classes—and helps us to understand their properties and how likely it is that an IPO belongs to a certain class. Specifically, LCA aims to identify latent classes of IPOs that share common traits and treats the sample as heterogeneous regarding the relationships between the involved variables. This means that the empirical analysis herein is not based on a theoretical IPO model derived from economic principles. To the best of our knowledge, this is the first study on the pricing of IPOs using LCA.

We identify four latent classes at each time horizon, where classes (i)-(ii) include a greater number of IPOs: (i) large-sized and underpriced IPOs; (ii) small-sized and overpriced IPOs; (iii) small-sized and severely underpriced IPOs; and (iv) large-sized IPOs that are overpriced at the one-day horizon but underpriced at the four-week horizon. The market returns are normal in latent classes (i)-(iii) and weak in class (iv). Thus, there is considerable heterogeneity in the data that would be hard to discover with traditional regression analysis. Moreover, approximately half of the IPOs in the technology sector are in the latent class with small-sized and overpriced IPOs, and most of the IPOs in the class with small-sized and severely underpriced IPOs are in the healthcare sector. Finally, the underpricing of IPOs is not corrected after one or four weeks of trading. Instead, the mean return and the standard deviation of returns increase with the time horizon.

The rest of this paper is organized as follows. The dataset is presented in the section Dataset and descriptive statistics and the pricing of IPOs is analyzed using LCA in the section Analyzing the pricing of IPOs using LCA. The section Discussion concludes the paper.

## Dataset and descriptive statistics

We examine a total of 314 IPOs in Denmark (31 IPOs), Finland (43 IPOs), Norway (65 IPOs) and Sweden (175 IPOs) issued by firms during the period of November 2009 through June 2019 on the following stock exchanges: Aktietorget, First North Copenhagen, First North Helsinki, First North Stockholm, Nordic MTF, OMX Copenhagen, OMX Helsinki, OMX Stockholm, Oslo Stock Exchange and Oslo Axess. It should be noted that the Nordic markets are highly integrated and to a large extent harmonized regarding the legal environments for listing and trading of securities [11, 12], which motivates our choice to pool the data when analyzing the pricing of IPOs using LCA.

The dataset includes information about the offer price for shares in the IPO, the first-day closing price, the first-week closing price, the four-week closing price, the number of shares offered in the IPO, the issuing firm's country of origin, the issue date, and the economic sector of the firm according to the Thomson Reuters Business Classification, where we used Thomson Reuters Datastream when collecting information on the IPOs. The information in the dataset was manually examined and crosschecked against various sources, such as brokerage firms and financial prospectuses.

First-day, first-week and four-week returns on shares in firm $i$ are calculated as

$$(1) \qquad \text{Return}_i^{\text{FD}} = \frac{\text{First} - \text{day price}_i - \text{Offer price}_i}{\text{Offer price}_i}$$

$$(2) \qquad \text{Return}_i^{\text{FW}} = \frac{\text{First} - \text{week price}_i - \text{Offer price}_i}{\text{Offer price}_i}$$

respective

$$(3) \qquad \text{Return}_i^{\text{4W}} = \frac{\text{Four} - \text{week price}_i - \text{Offer price}_i}{\text{Offer price}_i}$$

and the descriptive statistics of first-day, first-week and four-week returns are shown in Tables 1–3. For ease of comparison, the mean returns and the standard deviations of returns in Tables 1–3 have been converted to the four-week horizon in Table 4. Notably, but not unexpectedly, the magnitudes of the numbers in Table 4 are smaller for longer time horizons. (The figures in Tables 1 and 2 have been converted to the four-week horizon in A1 and A2 Tables in S1 Appendix).

The mean return and the standard deviation of returns are larger for longer time horizons. The mean returns at the different time horizons are 5.19, 5.98 and 7.77 percent, and the respective standard deviations of returns are 31.15, 33.51 and 40.89 percent. Hence, the average underpricing of IPOs is not corrected after one or four weeks of trading. Note that the mean returns are the equally weighted mean returns of the IPOs. The value weighted mean returns of the IPOs are calculated as well, with the offer sizes of the IPOs used as weights in the calculations.

Starting with first-day returns, three observations are made. First, the equally weighted mean return is negative in the first five years after the Great Recession and positive thereafter in the next six years. Thus, the IPOs are, on average, overpriced in the first years in the sample. This is unusual for IPOs. Nevertheless, as already highlighted, those five years are the first after the Great Recession, which might explain the unusual pattern in the pricing of IPOs. Furthermore, the number of IPOs issued during this period was relatively small; less than one-fourth of the IPOs in the sample were issued during 2009–2013.

**Table 1. Descriptive statistics of first-day returns on the issuing firms' securities.**

| Year | Observations | Min | Median | EW Mean | VW Mean | Max | SD |
|------|------|------|------|------|------|------|------|
| 2009 | 1 | -7.77% | -7.77% | -7.77% | -7.77% | -7.77% | - |
| 2010 | 31 | -69.70% | -1.96% | -0.42% | 6.50% | 120.45% | 40.63% |
| 2011 | 15 | -37.61% | -1.52% | -5.52% | -5.47% | 13.04% | 14.63% |
| 2012 | 8 | -63.19% | -2.72% | -4.58% | -2.83% | 30.51% | 26.33% |
| 2013 | 17 | -23.46% | 0.00% | -0.41% | 0.80% | 19.18% | 8.11% |
| 2014 | 33 | -78.97% | 2.92% | 0.38% | 9.95% | 34.58% | 23.96% |
| 2015 | 50 | -89.20% | 2.19% | 4.34% | 7.26% | 61.72% | 26.24% |
| 2016 | 40 | -88.22% | 3.91% | 9.91% | 0.95% | 108.90% | 35.46% |
| 2017 | 71 | -48.35% | 4.25% | 12.62% | 6.06% | 161.54% | 32.37% |
| 2018 | 38 | -54.55% | 0.67% | 3.48% | 5.51% | 208.09% | 39.23% |
| 2019 | 10 | -5.98% | 6.56% | 12.14% | 11.29% | 57.23% | 19.95% |
| All | 314 | -89.20% | 0.86% | 5.19% | 5.45% | 208.09% | 31.15% |

Note: *Observations* is the number of IPOs during a specific *Year*, *Min* is the minimum return, *Median* is the median return, *EW Mean* is the equally weighted mean return, *VW Mean* is the value weighted mean return, *Max* is the maximum return, and *SD* is the standard deviation of returns. The offer size of an IPO, which is the number of shares times the offer price for those shares, divided by the offer sizes of all IPOs in a given *Year* is used as the weight when calculating *VW Mean*.

Second, the value weighted mean return, in most years, is greater than the equally weighted mean return, suggesting that the first-day returns of large-sized IPOs are often higher than the corresponding returns of small-sized IPOs. In fact, since the value weighted mean returns in two of the first five years after the Great Recession are positive and not negative, as the equally weighted mean returns are, large-sized IPOs issued during those two years are underpriced. Additionally, the number of IPOs issued in those two years is greater than the number of IPOs issued in the other three of the first five years in the sample. Therefore, the unusual pattern in the pricing of IPOs in the first years after the Great Recession might be a small sample effect. Third, the standard deviation of returns is large throughout the sample period, reflecting the fact that there are IPOs with large positive first-day returns and IPOs with large negative returns.

**Table 2. Descriptive statistics of first-week returns on the issuing firms' securities.**

| Year | Observations | Min | Median | EW Mean | VW Mean | Max | SD |
|------|------|------|------|------|------|------|------|
| 2009 | 1 | -10.68% | -10.68% | -10.68% | -10.68% | -10.68% | - |
| 2010 | 31 | -68.18% | -3.55% | -3.77% | 6.24% | 86.36% | 35.41% |
| 2011 | 15 | -42.47% | -1.04% | -5.64% | -7.59% | 25.49% | 17.41% |
| 2012 | 8 | -8.85% | -2.11% | 14.42% | -6.39% | 122.03% | 44.10% |
| 2013 | 17 | -30.43% | -0.41% | -2.01% | 0.57% | 15.23% | 10.69% |
| 2014 | 33 | -78.77% | 6.07% | 3.61% | 9.44% | 45.02% | 20.23% |
| 2015 | 50 | -88.78% | 2.42% | 3.48% | 7.80% | 52.45% | 27.19% |
| 2016 | 40 | -89.07% | 3.83% | 9.94% | -0.50% | 102.74% | 35.09% |
| 2017 | 71 | -51.65% | 4.60% | 16.45% | 7.21% | 206.08% | 44.05% |
| 2018 | 38 | -59.09% | -0.19% | 0.86% | 7.78% | 152.21% | 32.61% |
| 2019 | 10 | -11.67% | 4.39% | 11.91% | 10.16% | 60.12% | 22.35% |
| All | 314 | -89.07% | 1.01% | 5.98% | 5.33% | 206.08% | 33.51% |

Note: *Observations* is the number of IPOs during a specific *Year*, *Min* is the minimum return, *Median* is the median return, *EW Mean* is the equally weighted mean return, *VW Mean* is the value weighted mean return, *Max* is the maximum return, and *SD* is the standard deviation of returns. The offer size of an IPO, which is the number of shares times the offer price for those shares, divided by the offer sizes of all IPOs in a given *Year* is used as the weight when calculating *VW Mean*.

**Table 3. Descriptive statistics of four-week returns on the issuing firms' securities.**

| Year | Observations | Min | Median | EW Mean | VW Mean | Max | SD |
|------|--------------|-----|--------|---------|---------|-----|-----|
| 2009 | 1 | -31.55% | -31.55% | -31.55% | -31.55% | -31.55% | - |
| 2010 | 31 | -71.97% | -1.54% | 1.52% | 12.61% | 102.27% | 35.24% |
| 2011 | 15 | -34.25% | -1.46% | -4.30% | -9.54% | 26.09% | 14.18% |
| 2012 | 8 | -65.96% | -1.51% | 18.12% | -9.96% | 233.90% | 90.21% |
| 2013 | 17 | -18.99% | -1.42% | -1.58% | 7.15% | 23.34% | 10.80% |
| 2014 | 33 | -79.11% | 5.48% | 5.71% | 11.18% | 121.10% | 29.38% |
| 2015 | 50 | -89.70% | 0.68% | 3.78% | 8.23% | 117.85% | 32.72% |
| 2016 | 40 | -89.36% | 4.85% | 12.82% | -2.63% | 260.27% | 50.03% |
| 2017 | 71 | -55.20% | 3.75% | 17.52% | 9.05% | 195.16% | 44.48% |
| 2018 | 38 | -68.18% | -1.75% | 1.56% | 12.85% | 217.39% | 46.37% |
| 2019 | 10 | -11.11% | 4.27% | 17.82% | 11.03% | 98.84% | 33.66% |
| All | 314 | -89.70% | 1.33% | 7.77% | 7.13% | 260.27% | 40.89% |

Note: *Observations* is the number of IPOs during a specific *Year*, *Min* is the minimum return, *Median* is the median return, *EW Mean* is the equally weighted mean return, *VW Mean* is the value weighted mean return, *Max* is the maximum return, and *SD* is the standard deviation of returns. The offer size of an IPO, which is the number of shares times the offer price for those shares, divided by the offer sizes of all IPOs in a given *Year* is used as the weight when calculating *VW Mean*.

Continuing to examine the first-week and four-week returns of the first five years after the Great Recession, the first-week returns are negative in four years, and the four-week returns are negative in three years. Moreover, there are no longer any striking differences between the equally weighted and value weighted mean returns. In fact, for the full sample period, the value weighted mean return is lower than the equally weighted mean return at the one-week and four-week horizons. Finally, as noted above, the standard deviations of first-week and four-week returns are even larger than the standard deviation of first-day returns.

Lastly, as also noted above, less than one-fourth of the IPOs were issued in the first five years in the sample (2009–2013), while more than three-fourth of the IPOs occurred in the

**Table 4. Descriptive statistics of first-day, first-week and four-week returns on the issuing firms' securities, where the mean returns and the standard deviations of returns have been transformed to the four-week horizon.**

| Year | First-day returns | | | First-week returns | | | Four-week returns | | |
|------|---------|---------|---------|---------|---------|---------|---------|---------|---------|
| | EW Mean | VW Mean | SD | EW Mean | VW Mean | SD | EW Mean | VW Mean | SD |
| 2009 | -80.15% | -80.15% | - | -36.35% | -36.35% | - | -31.55% | -31.55% | - |
| 2010 | -7.99% | 252.06% | 181.69% | -14.26% | 27.39% | 70.83% | 1.52% | 12.61% | 35.24% |
| 2011 | -67.90% | -67.53% | 65.41% | -20.74% | -27.09% | 34.81% | -4.30% | -9.54% | 14.18% |
| 2012 | -60.82% | -43.66% | 117.77% | 71.42% | -23.20% | 88.19% | 18.12% | -9.96% | 90.21% |
| 2013 | -7.83% | 17.23% | 36.25% | -7.80% | 2.29% | 21.38% | -1.58% | 7.15% | 10.80% |
| 2014 | 7.90% | 566.69% | 107.14% | 15.22% | 43.47% | 40.46% | 5.71% | 11.18% | 29.38% |
| 2015 | 133.72% | 306.34% | 117.36% | 14.65% | 35.04% | 54.38% | 3.78% | 8.23% | 32.72% |
| 2016 | 561.83% | 20.78% | 158.57% | 46.07% | -1.98% | 70.19% | 12.82% | -2.63% | 50.03% |
| 2017 | 976.98% | 224.52% | 144.75% | 83.89% | 32.11% | 88.10% | 17.52% | 9.05% | 44.48% |
| 2018 | 98.03% | 192.29% | 175.45% | 3.47% | 34.95% | 65.22% | 1.56% | 12.85% | 46.37% |
| 2019 | 889.14% | 749.29% | 89.21% | 56.84% | 47.25% | 44.69% | 17.82% | 11.03% | 33.66% |
| All | 174.84% | 188.77% | 139.32% | 26.17% | 23.09% | 67.02% | 7.77% | 7.13% | 40.89% |

Note: *EW Mean* is the equally weighted mean return, *VW Mean* is the value weighted mean return, and *SD* is the standard deviation of returns. The offer size of an IPO, which is the number of shares times the offer price for those shares, divided by the offer sizes of all IPOs in a given *Year* is used as the weight when calculating *VW Mean*.

final six years (2014–2019). This is not surprising because it is reasonable to believe that the Great Recession had a muting effect on firms wanting to go public. For example, Lowry and Schwert [13] found that firms tend to go public during periods characterized by large initial stock returns (cf. hot and cold IPO markets [14]).

## Analyzing the pricing of IPOs using LCA

In the empirical analysis, we identified four latent classes, numbered (i)-(iv), at each time horizon—that is, at the one-day, one-week and four-week horizons—in the pricing of IPOs in the Nordics. (AIC marginally decreased in value and BIC increased in value when a fifth latent class was added to the models.) See Tables 5–7 for the predicted latent class means using first-day, first-week and four-week returns on the issuing firms' securities.

First, there are two latent classes with large-sized IPOs (i.e., (i) and (iv)) and two classes with small-sized IPOs (i.e., (ii)-(iii)). Second, the market return on the issue dates of the IPOs is normal in the two latent classes with small-sized IPOs and in one class with large-sized IPOs (i.e., (i)-(iii)) but is weak in the other class with large-sized IPOs (i.e., (iv)). Third, the IPOs are overpriced in one latent class (i.e., (ii)) and underpriced in two classes (i.e., (i) and (iii)). In fact, the IPOs are severely underpriced in one latent class (i.e., (iii)). Fourth, the pricing of IPOs in one latent class shifts from being overpriced at the one-day horizon to being underpriced at the four-week horizon (i.e., (iv)). This latent class consists of roughly the same IPOs at each time horizon. See Table 8 for qualitative interpretations of the latent classes.

**Table 5. Predicted latent class means using first-day returns on the issuing firms' securities.**

|  |  | Margin | 95% CI |
|---|---|---|---|
| **Latent class (i)** | *First-Day Return* | 6.21% | [2.46%, 9.95%] |
|  | *Offer Size* | 1.95 | [1.84, 2.07] |
|  |  | 89.5 MUSD | [68.5 MUSD, 116.9 MUSD] |
|  | *Market Return* | -0.09% | [-0.24%, 0.06%] |
| **Latent class (ii)** | *First-Day Return* | -5.05% | [-10.67%, 0.56%] |
|  | *Offer Size* | 0.59 | [0.43, 0.74] |
|  |  | 3.9 MUSD | [2.7 MUSD, 5.5 MUSD] |
|  | *Market Return* | 0.08% | [-0.11%, 0.27%] |
| **Latent class (iii)** | *First-Day Return* | 124.84% | [107.15%, 142.52%] |
|  | *Offer Size* | 0.31 | [-0.06, 0.69] |
|  |  | 2.1 MUSD | [0.9 MUSD, 4.8 MUSD] |
|  | *Market Return* | 0.57% | [-0.06%, 1.20%] |
| **Latent class (iv)** | *First-Day Return* | -3.98% | [-32.57%, 24.62%] |
|  | *Offer Size* | 2.21 | [1.49, 2.92] |
|  |  | 161.1 MUSD | [31.1 MUSD, 834.3 MUSD] |
|  | *Market Return* | -3.19% | [-4.63%, -1.75%] |

Note: *First-Day Return* is the percentage change in the stock price after the first trading day compared with the offer price, *Offer Size* is the logarithm of the number of shares times the offer price for those shares in the IPO, where the numbers for *Offer Size* in the first row are expressed in million U.S. dollars (MUSD) in the second row, and *Market Return* is the stock market return in percent in the relevant market on the issue date of the IPO (i.e., the return on OMXC20 if the IPO's country of origin is Denmark, the return on OMXH25 if the IPO's country of origin is Finland, the return on OMXO20 if the IPO's country of origin is Norway, and the return on OMXS30 if the IPO's country of origin is Sweden). Margin is the marginal predicted latent class mean, and 95% CI is the 95 percent confidence interval for the marginal predicted latent class mean.

**Table 6. Predicted latent class means using first-week returns on the issuing firms' securities.**

| | | Margin | 95% CI |
|---|---|---|---|
| **Latent class (i)** | *First-Week Return* | 5.98% | [2.36%, 9.59%] |
| | *Offer Size* | 1.95 | [1.83, 2.06] |
| | | 88.4 MUSD | [67.5 MUSD, 115.8 MUSD] |
| | *Market Return* | -0.09% | [-0.24%, 0.06%] |
| **Latent class (ii)** | *First-Week Return* | -7.40% | [-13.27%, -1.53%] |
| | *Offer Size* | 0.58 | [0.43, 0.74] |
| | | 3.8 MUSD | [2.7 MUSD, 5.5 MUSD] |
| | *Market Return* | 0.09% | [-0.11%, 0.29%] |
| **Latent class (iii)** | *First-Week Return* | 119.36% | [104.11%, 134.60%] |
| | *Offer Size* | 0.36 | [0.05, 0.67] |
| | | 2.3 MUSD | [1.1 MUSD, 4.6 MUSD] |
| | *Market Return* | 0.32% | [-0.20%, 0.83%] |
| **Latent class (iv)** | *First-Week Return* | -0.10% | [-25.95%, 25.75%] |
| | *Offer Size* | 2.21 | [1.43, 2.99] |
| | | 162.0 MUSD | [27.1 MUSD, 966.5 MUSD] |
| | *Market Return* | -3.25% | [-4.68%, -1.81%] |

Note: *First-Week Return* is the percentage change in the stock price after the first trading week compared with the offer price, *Offer Size* is the logarithm of the number of shares times the offer price for those shares in the IPO, where the numbers for *Offer Size* in the first row are expressed in million U.S. dollars (MUSD) in the second row, and *Market Return* is the stock market return in percent in the relevant market on the issue date of the IPO (i.e., the return on OMXC20 if the IPO's country of origin is Denmark, the return on OMXH25 if the IPO's country of origin is Finland, the return on OMXO20 if the IPO's country of origin is Norway, and the return on OMXS30 if the IPO's country of origin is Sweden). Margin is the marginal predicted latent class mean, and 95% CI is the 95 percent confidence interval for the marginal predicted latent class mean.

How large are the latent classes at the different time horizons? See Table 9 for the predicted latent class probabilities using first-day, first-week and four-week returns on the issuing firms' securities. Two of the latent classes have a greater number of IPOs (i.e., (i)-(ii)), where the larger class consists of large-sized and underpriced IPOs issued when market returns are normal (i.e., (i)) and the other class consists of small-sized and overpriced IPOs also issued when market returns are normal (i.e., (ii)).

How many IPOs are in the latent classes? See Table 10 for the predicted number of IPOs in each latent class for different thresholds of the posterior probability using first-day, first-week and four-week returns on the issuing firms' securities. The tabulated numbers reflect what we learned in Table 9 about the sizes of the latent classes. Notably, although not shown in the table, most of the IPOs in the latent class with small-sized and severely underpriced IPOs are in the healthcare sector (i.e., (iii)). Moreover, approximately half of the IPOs in the technology sector are in the latent class with small-sized and overpriced IPOs (i.e., (ii)). The healthcare and technology sectors (66 and 70 IPOs, respectively) are the largest sectors in the sample.

Be aware that LCA does not divide the sample into mutually exclusive subsamples. On the one hand, if the bar is low for predicted latent class membership, then it is possible that a specific IPO may belong to more than one of the identified latent classes (see, e.g., when the posterior probability is equal to 0.1 in Table 10). On the other hand, if the bar is high for predicted latent class membership, then it is possible that a specific IPO does not belong to any of the identified latent classes (see, e.g., when the posterior probability is equal to 0.9 in Table 10).

**Table 7. Predicted latent class means using four-week returns on the issuing firms' securities.**

|  |  | **Margin** | **95% CI** |
|---|---|---|---|
| **Latent class (i)** | *Four-Week Return* | 4.86% | [0.65%, 9.08%] |
|  | *Offer Size* | 1.98 | [1.87, 2.08] |
|  |  | 94.6 MUSD | [74.1 MUSD, 120.7 MUSD] |
|  | *Market Return* | -0.08% | [-0.24%, 0.07%] |
| **Latent class (ii)** | *Four-Week Return* | -5.56% | [-11.52%, 0.39%] |
|  | *Offer Size* | 0.58 | [0.43, 0.73] |
|  |  | 3.8 MUSD | [2.7 MUSD, 5.4 MUSD] |
|  | *Market Return* | 0.12% | [-0.07%, 0.31%] |
| **Latent class (iii)** | *Four-Week Return* | 141.30% | [124.67%, 157.94%] |
|  | *Offer Size* | 0.59 | [0.32, 0.86] |
|  |  | 3.9 MUSD | [2.1 MUSD, 7.3 MUSD] |
|  | *Market Return* | -0.11% | [-0.59%, 0.36%] |
| **Latent class (iv)** | *Four-Week Return* | 6.14% | [-21.49%, 33.77%] |
|  | *Offer Size* | 2.21 | [1.46, 2.96] |
|  |  | 161.7 MUSD | [28.8 MUSD, 909.7 MUSD] |
|  | *Market Return* | -3.28% | [-4.66%, -1.90%] |

Note: *Four-Week Return* is the percentage change in the stock price after the first four trading weeks compared with the offer price, *Offer Size* is the logarithm of the number of shares times the offer price for those shares in the IPO, where the numbers for *Offer Size* in the first row are expressed in million U.S. dollars (MUSD) in the second row, and *Market Return* is the stock market return in percent in the relevant market on the issue date of the IPO (i.e., the return on OMXC20 if the IPO's country of origin is Denmark, the return on OMXH25 if the IPO's country of origin is Finland, the return on OMXO20 if the IPO's country of origin is Norway, and the return on OMXS30 if the IPO's country of origin is Sweden). Margin is the marginal predicted latent class mean, and 95% CI is the 95 percent confidence interval for the marginal predicted latent class mean.

**Table 8. Qualitative interpretations of the latent classes for first-day, first-week and four-week returns on the issuing firms' securities.**

|  |  | **Latent class (i)** | **Latent class (ii)** | **Latent class (iii)** | **Latent class (iv)** |
|---|---|---|---|---|---|
| **First-day returns** | *First-Day Return* | High | Low/negative | Very high | Low/negative |
|  | *Offer Size* | Large | Small | Small | Large |
|  | *Market Return* | Normal | Normal | Normal | Weak |
| **First-week returns** | *First-Week Return* | High | Low/negative | Very high | Normal |
|  | *Offer Size* | Large | Small | Small | Large |
|  | *Market Return* | Normal | Normal | Normal | Weak |
| **Four-week returns** | *Four-Week Return* | High | Low/negative | Very high | High |
|  | *Offer Size* | Large | Small | Small | Large |
|  | *Market Return* | Normal | Normal | Normal | Weak |

Note: *First-Day Return* is the percentage change in the stock price after the first trading day compared with the offer price, *First-Week Return* is the percentage change in the stock price after the first trading week compared with the offer price, *Four-Week Return* is the percentage change in the stock price after the first four trading weeks compared with the offer price, *Offer Size* is the logarithm of the number of shares times the offer price for those shares in the IPO, and *Market Return* is the stock market return in percent in the relevant market on the issue date of the IPO (i.e., the return on OMXC20 if the IPO's country of origin is Denmark, the return on OMXH25 if the IPO's country of origin is Finland, the return on OMXO20 if the IPO's country of origin is Norway, and the return on OMXS30 if the IPO's country of origin is Sweden). See Tables 5–7 for quantitative descriptions of the latent classes.

**Table 9. Predicted latent class probabilities using first-day, first-week and four-week returns on the issuing firms' securities.**

|  |  | Margin | 95% CI |
|---|---|---|---|
| **First-day returns** | **Latent class (i)** | 0.608 | [0.520, 0.689] |
|  | **Latent class (ii)** | 0.351 | [0.274, 0.437] |
|  | **Latent class (iii)** | 0.026 | [0.013, 0.051] |
|  | **Latent class (iv)** | 0.015 | [0.003, 0.066] |
| **First-week returns** | **Latent class (i)** | 0.613 | [0.525, 0.695] |
|  | **Latent class (ii)** | 0.332 | [0.256, 0.419] |
|  | **Latent class (iii)** | 0.040 | [0.023, 0.070] |
|  | **Latent class (iv)** | 0.014 | [0.003, 0.065] |
| **Four-week returns** | **Latent class (i)** | 0.595 | [0.513, 0.671] |
|  | **Latent class (ii)** | 0.344 | [0.272, 0.424] |
|  | **Latent class (iii)** | 0.047 | [0.028, 0.079] |
|  | **Latent class (iv)** | 0.014 | [0.003, 0.061] |

Note: Margin is the marginal predicted latent class mean, and 95% CI is the 95 percent confidence interval for the marginal predicted latent class mean. See Tables 5–7 for quantitative descriptions of the latent classes, and see Table 8 for qualitative interpretations of the latent classes.

We also asked ourselves the extent to which the offer size of IPOs and the market return on the issue date of IPOs explain their under- or overpricing. To answer this question, we ran several least squares regressions with the first-day, first-week respective four-week returns as the dependent variables. Fixed effects accounting for the IPO's country of origin, its economic sector classification, and the year of its issue date were added to the regressions. See Table 11 for estimation results.

We found that neither the offer size of IPOs nor the market return on their issue date had a significant effect on their under- or overpricing in the regressions. Apparently, there is not a linear relationship between these variables. At the same time, it is probable that the regressions suffer from omitted variable bias. For instance, variables for firm characteristics (e.g., earnings management, firm age, firm size, leverage), IPO characteristics (e.g., underwriter quality) and market sentiment (e.g., number of IPOs occurring shortly before the focal IPO) might have

**Table 10. Predicted number of IPOs in each latent class for different thresholds of the posterior probability using first-day, first-week and four-week returns on the issuing firms' securities.**

|  | Probability | Latent class (i) | Latent class (ii) | Latent class (iii) | Latent class (iv) |
|---|---|---|---|---|---|
| **First-day returns** | 0.1 | 238 | 160 | 8 | 8 |
|  | 0.5 | 195 | 106 | 8 | 5 |
|  | 0.9 | 137 | 67 | 8 | 1 |
| **First-week returns** | 0.1 | 235 | 151 | 14 | 7 |
|  | 0.5 | 196 | 100 | 12 | 5 |
|  | 0.9 | 142 | 66 | 12 | 1 |
| **Four-week returns** | 0.1 | 226 | 153 | 16 | 6 |
|  | 0.5 | 189 | 104 | 15 | 5 |
|  | 0.9 | 140 | 72 | 13 | 1 |

Note: Probability is the threshold of the posterior probability. See Tables 5–7 for quantitative descriptions of the latent classes, and see Table 8 for qualitative interpretations of the latent classes.

**Table 11. Least squares regressions.**

| First-day returns | Constant | 12.177* (0.076) | 12.685* (0.088) | 12.434* (0.072) | 12.741* (0.090) |
|---|---|---|---|---|---|
| | Offer Size | | -0.420 (0.876) | | -0.257 (0.923) |
| | Market Return | | | 1.274 (0.503) | 1.254 (0.499) |
| | Observations | 314 | 314 | 314 | 314 |
| | R-square | 0.088 | 0.088 | 0.089 | 0.089 |
| First-week returns | Constant | 18.865** (0.014) | 19.088** (0.022) | 18.778** (0.014) | 19.068** (0.022) |
| | Offer Size | | -0.185 (0.943) | | -0.243 (0.925) |
| | Market Return | | | -0.429 (0.813) | -0.448 (0.803) |
| | Observations | 314 | 314 | 314 | 314 |
| | R-square | 0.106 | 0.106 | 0.106 | 0.106 |
| Four-week returns | Constant | 20.301* (0.058) | 22.600** (0.050) | 19.859* (0.061) | 22.495** (0.048) |
| | Offer Size | | -1.899 (0.550) | | -2.205 (0.493) |
| | Market Return | | | -2.185 (0.262) | -2.356 (0.231) |
| | Observations | 314 | 314 | 314 | 314 |
| | R-square | 0.093 | 0.094 | 0.095 | 0.097 |

Note: The dependent variable in the least squares regressions with robust standard errors is the first-day return, the first-week return respective the four-week return on the issuing firms' securities. *Offer Size* is the logarithm of the number of shares times the offer price for those shares in the IPO, and *Market Return* is the stock market return in percent in the relevant market on the issue date of the IPO (i.e., the return on OMXC20 if the IPO's country of origin is Denmark, the return on OMXH25 if the IPO's country of origin is Finland, the return on OMXO20 if the IPO's country of origin is Norway, and the return on OMXS30 if the IPO's country of origin is Sweden). Dummy variables for year, with 2019 as the baseline category, the IPO's country of origin, with Sweden as the baseline category, and the economic sector classification of the IPO according to the Thomson Reuters Business Classification, with the technology sector as the baseline category, are included in the regressions. *Observations* is the number of IPOs in a regression, and *R-square* is the fraction of the dependent variable that is explained by the model. *p*-values are in parentheses. Significance levels: $^*p<0.1$, $^{**}p<0.05$ and $^{***}p<0.01$.

explanatory power for the under- or overpricing of IPOs. However, this is not the point here. The advantage of using LCA is that this structural equation methodology is able to detect patterns in sparse and heterogeneous data that ordinary regression analysis might not uncover.

## Discussion

Three results stand out from this study of the underpricing of 314 IPOs in the Nordics after the Great Recession. First, the underpricing of IPOs is not corrected after one or four weeks of trading. Instead, the mean return and the standard deviation of returns increase with the time horizon.

Second, four latent classes were identified at each time horizon, with classes (i)-(ii) including a greater number of IPOs: (i) large-sized and underpriced IPOs; (ii) small-sized and overpriced IPOs; (iii) small-sized and severely underpriced IPOs; and (iv) large-sized IPOs that are overpriced at the one-day horizon but underpriced at the four-week horizon. The market returns are normal in the first three latent classes and weak in the fourth. Third, approximately

half of the IPOs in the technology sector are in the latent class with small-sized and overpriced IPOs, and most of the IPOs in the class with small-sized and severely underpriced IPOs are in the healthcare sector.

What is the value-added of using LCA when studying IPOs? LCA treats the sample with IPOs as heterogeneous regarding the relationships between the involved variables. This should be contrasted with ordinary regression analysis, which assumes that the dependent and explanatory variables behave uniformly over the whole sample. In other words, LCA is able to identify more than one group—or latent class—of IPOs that share common traits. For example, LCA revealed in this study that there are four latent classes of IPOs in the Nordics with different qualitative properties. Hence, LCA is able to detect patterns in a sample that ordinary regression analysis might miss. For this reason, LCA is a valuable complement to traditional regression analysis when studying IPOs.

## Supporting information

**S1 Data.**
(ZIP)

**S1 Appendix.**
(DOCX)

## Acknowledgments

This paper has benefited from comments by Joakim Westerholm and two anonymous reviewers. All errors are entirely our own.

## Author Contributions

**Formal analysis:** Mikael Bask, Anton Läck Nätter.

**Writing – original draft:** Mikael Bask.

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
