## [Decision Letter · Decision Letter 0]

14 Jun 2021

PONE-D-21-08567

Market Return, Offer Size and the Underpricing of IPOs in the Nordics

PLOS ONE

Dear Dr. Bask,

Thank you for submitting your manuscript to PLOS ONE. After careful consideration, I feel that it has merit but does not fully meet PLOS ONE’s publication criteria as it currently stands. 

The reviewers consider that the paper fails in several relevant points that seriously difficult its publication.

The major concern is that the theoretical basis is not clear. As one of the reviewers remark the mere fact that Nordic IPO markets are under-analysed result a poor motivation. I agree with the reviewers that the pricing of IPOs is a wide investigated research topic in the finance literature, and I find also that it is not clear what is the contribution of this paper to the literature on this topic. In this line, the literature review is far from being exhaustive and consequently needs to be extended.

Another concern that needs clarification is the sample used and the relevance of focusing in 4 countries.

Concerns showed by the reviewers are enough to reject this manuscript, but I also consider that comments can substantially enrich this research. Therefore, I invite you to submit a revised version of the manuscript that addresses the points raised during the review process.

We look forward to receiving your revised manuscript.

Kind regards,

J E. Trinidad Segovia

Academic Editor

PLOS ONE

Journal Requirements:

Reviewers' comments:

Reviewer's Responses to Questions

**Comments to the Author**

1. Is the manuscript technically sound, and do the data support the conclusions?

Reviewer #1: Yes

Reviewer #2: Yes

Reviewer #3: Yes

2. Has the statistical analysis been performed appropriately and rigorously? 

Reviewer #1: Yes

Reviewer #2: Yes

Reviewer #3: Yes

3. Have the authors made all data underlying the findings in their manuscript fully available?

Reviewer #1: Yes

Reviewer #2: Yes

Reviewer #3: No

4. Is the manuscript presented in an intelligible fashion and written in standard English?

Reviewer #1: Yes

Reviewer #2: Yes

Reviewer #3: Yes

5. Review Comments to the Author

Reviewer #1: This paper explores whether the average underpricing, or overpricing, of IPOs is corrected after one or four weeks of trading, and examines the relationship between offer size of IPOs or the market return on their issue date and their underpricing. The paper is based on proper data and methods, but there are several issues which should be revised.

1. The theoretical basis of the paper is relatively weak, and there are too few references to the literature. It is better to expand the research content on the basis of the existing research. It is suggested to increase the research basis of the core issues of the paper. For example, this paper points out that the reason why there is few studies on the Nordic IPO market is that the market is small, which is not convincing. In addition to listing the relevant data, it also needs to sort out and summarize the relevant literature of the Nordic IPO market, so as to further put forward the research content.

2. Add the research contribution of the paper in the section of “1. Introduction”.

3. The formula for calculating returns on shares in "3. Descriptive statistics" is more appropriate in the section of "2. Dataset ". In the "2. Dataset" section, sample selection and data sources should be introduced, and the definition and measurement methods of the main variables in the text should be accurately stated, and a table about the definition and description of the variables should be listed.

4. Construct the regression model of the relationship between the offer size of IPOs or the market return on their issue date and IPO pricing in the paper. And in the "2. Dataset" section”, the variables involved in the model are defined in detail to increase the rationality of the paper structure and present the main content more clearly.

5. This paper found that neither the offer size of IPOs nor the market return on their issue date had a significant effect on their underpricing. In the interpretation of the results, only the possible deviations of missing variables are mentioned. It is recommended to add the specific reasons for this result, otherwise the test in this part will be meaningless.

Reviewer #2: please also see attached file if available (formatted same content)

Referee report for manuscript PONE-D-21-08567 i

Thank you for submitting an interesting paper.

The paper is updating research on the important market for Nordic IPOs, not because of its size but because of that I has been a breeding ground for important innovation. It is also a market with relatively strict regulatory requirements on new listings in international comparison and hence can serve as a best practice ground, see Westerholm (2007). The paper by Bask and Lack Natter looks at all markets in the region which I think is a good approach as they are economically interconnected and comparable in terms of regulation, also their stock exchanges are either merged or co-listing each other’s shares. The paper finds that

I have a few suggestions how to improve the final version of the paper

1. line 38 whether the average underpricing, or overpricing, of IPOs is corrected after one or four weeks 38 of trading. This is a dimension in the pricing of IPOs that is under-researched.

I would propose say: “Intermediate and long-run underperformance is relatively under-researched” AND footnote some of the papers that do look at long run underperformance e.g. Pseudo Market Timing and the Long-Run Underperformance of IPOs by Paul Shultz (2003), check google scholar for later referenced to this paper. Westerholm (2007) also reports intermediate to long run returns.

2. Line 59 The rest of the note is organized as follows – Should you use “paper” rather than “note” in my view this is a full paper not a note on previous work?

3. The paper examines a total of 314 IPOs in Denmark (31 IPOs), Finland (43 IPOs), Norway (65 IPOs) and Sweden 63 (175 IPOs) issued by firms during the period of November 2009 through June 2019. The number seems a little low do you only include major list companies?

4. Empirical analysis You compare returns and standard deviations for different horizons.

4.1 I suggest you need to consider a robustness check when you compute excess returns for the IPO stock corrected for a market index during the same period. Attention you do already report the market index and show they don’t move much during 1 or 4 weeks so I am happy with that, if you wanted you could compute the difference between stock returns and index returns for the respective periods.

4.2 You should at the very least annualise the returns so that they are comparable across different horizons. (You can annualise the 1 week standard deviation by multiplying by square root out of 52). If you prefer not to annualise as these return may look large, transform them all to one month returns and compare. This can be separate appendix table only comparing this for table 1 and 4.

5. Conclusions: think of something general we can learn from your analysis with regards to market efficiency, of important to investors, or regulators.

Thank you for the opportunity to read your work and best of luck with the revision of the final version.

I do not need to see these corrections and I will recommend an acceptance of your paper.

Best wishes

Referee

Reviewer #3: This manuscript aims at addressing the interaction between offer size and underpricing of IPOs for a sample of 314 firms from 4 Nordic countries.

The pricing of IPOs is a heavily investigated research topic in the finance literature. It is unclear what this manuscript adds to our understanding of IPO pricing:

1. A recent meta study on IPO pricing [Engelen, P. J., Heugens, P., Van Essen, M., Turturea, R., & Bailey, N. (2020). The impact of stakeholders’ temporal orientaton on short-and long-term IPO outcomes: A meta-analysis. Long Range Planning, 53(2), 101853.] addresses the pricing of IPOs according to those two dimensions (proceeds and underpricing). The authors should embed their manuscript within the IPO literature and clearly demonstrate their contribution. What do we learn from the current study which was not addressed in this meta-study?

2. In what way does the focus on 4 Nordic countries help us to better understand IPO underpricing. The mere fact that Nordic IPO markets are under-analyzed is a poor motivation for the study. There are several cross-country studies in the IPO literature, including the Nordic countries. So what make the focus on those 4 countries so particular that it helps us to develop our understanding of the IPO phenomenon?

3. The extension of the time horizon from one day, to one week and to four weeks has been done several times in the IPO literature as well and did not show any spectacular additional insights. It is unclear in what way the current study would change that conclusion.

4. The sample size is relatively small, about 300 IPOs, so distributing this over 4 groups renders very small subsample sizes, questioning the relevance of the results.

5. In what why is the institutional framework of Denmark, Finland, Norway and Sweden alike or different? Can we just pool all IPOs together?

6. PLOS authors have the option to publish the peer review history of their article (what does this mean?). If published, this will include your full peer review and any attached files.

Reviewer #1: No

Reviewer #2: **Yes: **Joakim Westerholm

Reviewer #3: No

---

## [Author Response · Author response to Decision Letter 0]

29 Jul 2021

Please see the attached letters to the reviewers.

---

## [Decision Letter · Decision Letter 1]

31 Aug 2021

PONE-D-21-08567R1

Latent class analysis of IPOs in the Nordics

PLOS ONE

Dear Dr. Bask,

My first decision about this work was a major revision even considering that one of the reviewers had serious doubts about some aspects of this research.

After a second round and considering that both reviewers accept to revise this new version, I afraid that authors have not been able to provide a convincing answer to some of the major concerns, such us, what is the contribution of this research to the existing literature on this topic. Therefore, after careful consideration, I have decided that your manuscript does not meet our criteria for publication and must therefore be rejected.

I am sorry that we cannot be more positive on this occasion, but hope that you appreciate the reasons for this decision.

Yours sincerely,

J E. Trinidad Segovia

Academic Editor

PLOS ONE

Reviewers' comments:

Reviewer's Responses to Questions

**Comments to the Author**

1. If the authors have adequately addressed your comments raised in a previous round of review and you feel that this manuscript is now acceptable for publication, you may indicate that here to bypass the “Comments to the Author” section, enter your conflict of interest statement in the “Confidential to Editor” section, and submit your "Accept" recommendation.

Reviewer #1: (No Response)

Reviewer #3: (No Response)

2. Is the manuscript technically sound, and do the data support the conclusions?

Reviewer #1: (No Response)

Reviewer #3: Yes

3. Has the statistical analysis been performed appropriately and rigorously? 

Reviewer #1: (No Response)

Reviewer #3: Yes

4. Have the authors made all data underlying the findings in their manuscript fully available?

Reviewer #1: (No Response)

Reviewer #3: No

5. Is the manuscript presented in an intelligible fashion and written in standard English?

Reviewer #1: (No Response)

Reviewer #3: Yes

6. Review Comments to the Author

Reviewer #1: (No Response)

Reviewer #3: The author made some good faith efforts in clarifying some issues raised by the reviewers. Among other, the authors now clarify that their main contribution to the literature is the use of latent class analysis (LCA) as the statistical tool when analyzing IPO data. The author team also adjusted the title of their manuscript to reflect this.

Although the aimed contribution of the manuscript is now more clear, it also opens up the issue of the significant contribution of the paper. Just applying another methodology to a well-researched topic, does not constitute a contribution in itself. Only when the new method advances our understanding of the field, this warrants publication. I still fail to see the contribution of the manuscript.

Many of my concerns still stand:

1. The authors did not clearly demonstrate what we can learn from the current study which was not addressed in the meta-study. Apparently the meta study did not include any Nordic countries. Fine, but apart from this observation, what do we learn from the new study that we did not know before? What did LCA learns us that we did not know before on IPOs?

2. In what way does the focus on 4 Nordic countries help us to better understand IPO underpricing? The author argue that their contribution to the literature is found in the method used—latent class analysis (LCA)—and not in any particular difference between the Nordic IPO market and other (small) IPO markets. This bring us again to point 1.

3. The authors acknowledge that the extension of the time horizon from one day, to one week and to four weeks is not any new insight, the main contribution is again LCA.

4. Thank you for the clarification.

5. In what way is the institutional framework of Denmark, Finland, Norway and Sweden alike or different? Can we just pool all IPOs together? The author did not address any differences or similarities in institutional frameworks of those Nordic countries. They merely acknowledge pooling all IPOs together without any argumentation whether this is appropriate or not.

Overall, I fail to see the contribution of LCA to the IPO literature.

7. PLOS authors have the option to publish the peer review history of their article (what does this mean?). If published, this will include your full peer review and any attached files.

Reviewer #1: No

Reviewer #3: No

- - - - -

---

## [Author Response · Author response to Decision Letter 1]

22 Sep 2021

Please, see the attached appeal letter in which we addresses the concerns by the Academic Editor and the reviewer.

---

## [Editor Report · Decision Letter 2]

21 Oct 2021

Latent class analysis of IPOs in the Nordics

PONE-D-21-08567R2

Dear Dr. Bask,

We’re pleased to inform you that your manuscript has been judged scientifically suitable for publication and will be formally accepted for publication once it meets all outstanding technical requirements.

Kind regards,

Maurizio Naldi

Academic Editor

PLOS ONE
---

## [Editor Report · Acceptance letter]

25 Oct 2021

PONE-D-21-08567R2 

Latent class analysis of IPOs in the Nordics 

Dear Dr. Bask:

I'm pleased to inform you that your manuscript has been deemed suitable for publication in PLOS ONE. Congratulations! Your manuscript is now with our production department. 

Kind regards, 

on behalf of

Professor Maurizio Naldi 

Academic Editor

PLOS ONE